# Calcite as a Precursor of Hydroxyapatite in the Early Biomineralization of Differentiating Human Bone-Marrow Mesenchymal Stem Cells

**DOI:** 10.3390/ijms22094939

**Published:** 2021-05-06

**Authors:** Andrea Sorrentino, Emil Malucelli, Francesca Rossi, Concettina Cappadone, Giovanna Farruggia, Claudia Moscheni, Ana J. Perez-Berna, Jose Javier Conesa, Chiara Colletti, Norberto Roveri, Eva Pereiro, Stefano Iotti

**Affiliations:** 1Mistral Beamline, ALBA Synchrotron Light Source, Cerdanyola del Valles, 08290 Barcelona, Spain; asorrentino@cells.es (A.S.); anperez@cells.es (A.J.P.-B.); jj.conesa@cnb.csic.es (J.J.C.); epereiro@cells.es (E.P.); 2Department of Pharmacy and Biotechnology, University of Bologna, 40127 Bologna, Italy; francesca.rossi105@unibo.it (F.R.); concettina.cappadone@unibo.it (C.C.); giovanna.farruggia@unibo.it (G.F.); stefano.iotti@unibo.it (S.I.); 3National Institute of Biostructures and Biosystems, 00136 Rome, Italy; 4Department of Biomedical and Clinical Sciences “Luigi Sacco”, Università degli Studi di Milano, 20157 Milan, Italy; claudia.moscheni@unimi.it; 5Chemical Center S.r.l, Granarolo dell’ Emilia, 40057 Bologna, Italy; collettichiara8@gmail.com (C.C.); roveri.norberto@gmail.com (N.R.)

**Keywords:** osteoblastic differentiation, biomineralization, bone marrow mesenchymal stem cells, cryo-XANES

## Abstract

Biomineralization is the process by which living organisms generate organized mineral crystals. In human cells, this phenomenon culminates with the formation of hydroxyapatite, which is a naturally occurring mineral form of calcium apatite. The mechanism that explains the genesis within the cell and the propagation of the mineral in the extracellular matrix still remains largely unexplained, and its characterization is highly controversial, especially in humans. In fact, up to now, biomineralization core knowledge has been provided by investigations on the advanced phases of this process. In this study, we characterize the contents of calcium depositions in human bone mesenchymal stem cells exposed to an osteogenic cocktail for 4 and 10 days using synchrotron-based cryo-soft-X-ray tomography and cryo-XANES microscopy. The reported results suggest crystalline calcite as a precursor of hydroxyapatite depositions within the cells in the biomineralization process. In particular, both calcite and hydroxyapatite were detected within the cell during the early phase of osteogenic differentiation. This striking finding may redefine most of the biomineralization models published so far, taking into account that they have been formulated using murine samples while studies in human cell lines are still scarce.

## 1. Introduction

Biomineralization (BM) is the process by which organisms form organized mineral crystals during bone formation. During the BM process, ions are converted to biominerals thanks to chemical–physical transformations performed by the cellular activity. This elaborate process, still not completely elucidated, creates composite biomaterials made of organic and inorganic compounds, with a complex architecture. Indeed, it is known that bone, as other biomineralized tissues [1], shows a complex structure organized at multiple length scales, from the molecular to the macroscopic level [2]. On the nanometric scale, the highly specialized organic matrix of collagen microfibrils seems to direct the formation of nanosized platelet–hydroxyapatite (HA) oriented in parallel to the collagen fibril axis [3,4,5]. Together with the collagenous matrix, various highly acidic non-collagenous proteins are believed to strongly affect the mineralization process by binding to the surface of mineral particles [6]. Alongside these organic components, amorphous calcium phosphate is commonly considered the mineral precursor in the growth of HA nanocrystals. However, the process that explains the intracellular genesis and the consecutive propagation in the extracellular matrix of the biogenic mineral remains largely unexplained.

At the cellular level, the formation, maintenance, and repair of bone are based on the complex crosstalk between osteoclasts, osteoblasts, and osteocytes [7]. Osteoblasts differentiate from bone mesenchymal stem cells (bMSC) and promote bone deposition [8]. Osteoclasts develop from hematopoietic progenitors and promote bone re-sorption [7]. As the precursor of osteoblasts and osteocytes, bMSCs are heavily investigated to explore the molecular basis of osteogenesis with the aim to translating basic knowledge to application in tissue engineering and regenerative medicine [8]. Calcium (Ca) is a key element for bone, and it is present in the extracellular mineralized matrix as an integral component of HA crystals. It is noteworthy that a deficiency of extracellular Ca not only alters bone mass but also affects the behavior of bMSC [9]. The knowledge about both the genesis within the cells and the evolution of the extracellular different Ca-phosphates and polyphosphates compounds in the process of HA deposition is still incomplete. In fact, core knowledge is provided by studies on the advanced phases of biomineralization [4,5,10].

To tackle the challenge of characterizing such a complex and fascinating phenomenon in its early stages, we exploited the remarkable versatility of synchrotron-based cryo-soft-X-rays transmission microscopy techniques. In this work, the localization and characterization of both crystalline phase and Ca concentration in the Ca-rich depositions at the early phase of biomineralization in human bMSC were performed using cryo-soft-X-ray tomography (cryoSXT) and cryo-XANES microscopy (cryoXANES). CryoSXT consist of recording 2D soft X-ray transmission projections on a CCD detector changing the sample orientation with respect the incident photons. Then, 3D maps of the sample are generated from the angular series by applying a reconstruction algorithm. Scanning the energy across the edge of interest, cryoXANES provides pixel by pixel absorption spectra and can be exploited to determine the 2D chemical state of both light and heavy elements in the sample. The combination of these techniques permits to study biomineralization at the intracellular level in frozen hydrated whole cells at native state condition and with a spatial resolution of few tens of nanometers [11,12,13]. Three whole cells per each condition (bMSC at 4 and 10 days after osteoblastic induction) for a total number of 107 Ca depositions were examined. The results show that crystalline calcite (the most stable polymorph of CaCO_3_) is present in Ca-rich depositions in bMSC at 4 days after osteoblastic induction (bMSC-4D), while HA crystals and no CaCO_3_ depositions were found in bMSC at 10 days after osteoblastic induction (bMSC-10D). This suggests that calcite is one of the precursors in the biomineralization process, which culminates with the formation of HA in bMSC.

## 2. Results

### 2.1. 3D Cell Ultrastructure and 2D Ca Intracellular Distribution

Synchrotron-based cryoSXT was used for the 3D imaging of cryo-preserved bMSC-4D and bMSC-10D with a spatial resolution in the range of tens of nanometers [14]. bMSC were plunged frozen in liquid ethane and then imaged unfixed, unstained, and unsectioned in a quasi-native state. Angular series were acquired using a photon energy of 520 eV, where the water-rich cytoplasm is almost transparent with respect to dense organic and inorganic intracellular structures, which will show a higher (lower) absorption (transmission) contrast. A bMSC-4D 3D central slice reconstruction, constituted by four stitched tomograms, is shown in Figure 1a. It displays the nucleus at the center and the surrounding cytoplasm showing vacuoles, a variety of vesicles, cytoskeleton, and other cellular structures commonly visible in TEM resin-embedded thin sections [15]. The automatic segmentation of all the dense objects, which are abundant in this kind of cell, is reported in Figure 1b,c for bMSC-4D. As shown in Figure 2, nucleus (N), mitochondria (green arrows), many vacuoles (V1, V2, and V3) and dense bodies (D) are recognizable in both X-ray and electron microscopy data (Figure 2a,b, respectively). Mitochondria appear dense and well conserved. Multilamellar dense bodies are found isolated or in contact with a vesicle. Interestingly, we can distinguish different kinds of vesicles depending on the transmission contrast: low absorbing isolated vesicles (V1), vesicles in contact or containing a dense structure (V2), and vesicles containing smaller dense objects (V3). The latter are visible only in the TEM images, either because of the highest spatial resolution or the different sample preparation, which requires the cell dehydration and then a possible clustering of organic materials.

To distinguish Ca-rich bodies from carbon dense structures such as lipid droplets, liposomes, or dense membranes, the energy of the incident photons was varied across the Ca L edges. In Figure 3a,b, an ROI in the cytoplasm of the same cell shown in Figure 1 is imaged at the Ca L_3_ pre-edge (≈342 eV) and at the L_3_ peak maxima (≈349.2 eV), where Ca atoms are strongly absorbing the incoming radiation. Since the absorption of all the other elements is nearly the same at these two energies, the differential image reported in Figure 3c is gives the Ca 2D distribution in the field of view.

In order to quantitatively define the 2D Ca distribution, a signal-to-noise ratio (S/N) criterion was used between the Ca pre-edge and the L_3_ peak maxima absorbance value. As explained in detail in the methods section, only pixels with $∆\mu_{l}t=\left[\mu_{l}\left(349.2eV\right)-\mu_{l}\left(342eV\right)\right]t>2N$, with *µ_l_* linear absorption coefficient, *t* thickness, and N noise in the pre-edge energy region were considered to contain Ca. This method allows defining the 2D Ca distribution in the field of view and also combining the 2D information with cryoSXT to quantify the Ca concentration and to distinguish the Ca-rich body in the 3D reconstructed volume, as shown in Figure 3d,e for another bMSC-4D.

Ca distribution maps show that pixels containing Ca are grouped in clusters of different dimensions corresponding to different Ca depositions. All the dense bodies localized inside or in proximity of big vesicles and clearly detectable in both the cryoSXT reconstruction and in the TEM section images, as reported in the precedent paragraph (see Figure 2), show no Ca contents by spectroscopy analysis. The presence of Ca depositions at both 4 and 10 days after osteoblastic induction raises questions regarding their chemical compositions, their degree of crystallinity, their concentrations, and the possible correlation of these quantities with the depositions dimensions.

### 2.2. Chemical Fingerprint of Ca Depositions

The Ca L_3,2_-edges XANES of the calcium carbonates and phosphates consists of two main spin-orbit related peaks, L_3_ and L_2_, along with a number of smaller peaks that appear to precede the L_3_ and L_2_ main peaks. The origin of this multi-peaks pattern is known to be the crystal field arising from the symmetry of the atoms surrounding the Ca^2+^ ion in the first coordination sphere [16]. Therefore, Ca carbonates and Ca phosphates, in their crystalline forms, have different spectrum near edge structures because of the different short-range order around the Ca ions in the two cases [17,18]. These features allow distinguishing the carbonates from phosphate quite easily, even with a moderate S/N and spectral resolution (≈0.2 eV). 

Therefore, we performed cryoXANES on selected bMSC-4 and bMSC-10 with the purpose of defining the Ca chemical state in the detected Ca-rich depositions. The typical spectrum from bMSC-4 depositions is reported in Figure 4a and is characterized by strong L_3_ and L_2_ pre-peaks (a_1_ and b_1_ Figure 4b) and closely resembles the calcite spectrum, as it is demonstrated by the comparison with a measured calcite reference spectrum (reported in Figure 4b). All the Ca depositions identified in bMSC-10 show a similar absorption spectrum with spectral features in the multi-peaks pattern region that can be attributed to the presence of HA crystals. An HA reference spectra is reported in Figure 4c. The small intensity peaks structure before the L_3_ main peak (1, 2, a_1_, a_0_) and the peculiar “hook” shape of the L_2_ pre-peak (b_1_), which is the convolution of three small peaks [17,19], are well known HA Ca-L edge absorption spectrum features. The presence of HA crystals inside bMSC-10 was already reported in the work by Procopio and colleagues [20]. bMSC-4D spectra from the field of view of Figure 3a–c is reported as an example in Figure 5, both with the binary map showing the Ca depositions (Figure 5a), i.e., the pixels that satisfy the S/N criteria defined in the precedent paragraph. The binary map highlights six depositions, and the respective calcite spectra are shown in Figure 5b–g. Interestingly, in a few cases, spectra similar to the one of HA were also extracted from depositions in bMSC-4D. These few specific spectra are reported in Figure 6. Only one of these rare HA depositions was in addition characterized with tomography at the Ca L edge, and its volume and the corresponding estimated Ca concentration are indicated with a blue data point in Figure 7b,c. Even if we cannot exclude the co-localization in both bMSC-4 and bMSC-10 Ca-rich objects of relatively small quantities of other Ca carbonates and phosphates phases, these observations demonstrate that Ca carbonates and in particular calcite is one of the mineral precursors of HA in bMSC, as we will show with more detail in the next paragraph.

### 2.3. Ca Carbonate Phase

It is well known that the ratio of the L_2_ pre-peak (L2′) to the L_2_ peak (L2′/L2) is a good indication of the short-range order of the oxygen atoms around the Ca ions [11,21,22,23] in Ca carbonates compounds. In Figure 7a, the calculated value of this ratio is reported for the Ca depositions in bMSC-4D, which shows a spectrum that can be attributed to some form of Ca carbonate. The details of the calculation are reported in the methods section. The red dashed line is the corresponding value measured on a calcite reference spectrum. Measured values from depositions and from the reference are compatible within the error. This demonstrates that most of the Ca bMSC-4D depositions are crystalline calcite.

### 2.4. Depositions Volume

The measure of the Ca depositions volume was achieved by cryoSXT at 349.2 eV, i.e., on the Ca L_3_ resonance where Ca is strongly absorbing and dominates the contrast in the image. Ca depositions can be easily selected with a local threshold. The well-known elongation on the z-direction due to the missing wedge (projections were collected between −55 deg and +55 deg) was calibrated on the spherical gold beads deposited on the grid just before sample freezing. Additionally, to this missing wedge correction factor, beads were also used to simplify the alignment of the projections on a common rotation axis. The used formula for the volume *V* was V=kv, with *k* number of selected voxels by the local threshold and *v* voxel dimension, ((13 nm)^3^). The spatial resolution, using the Fourier Shell Correlation method [24], was estimated to be about 58 nm half pitch corresponding to about 4.5 pixels. Then, the corresponding volume error bar was calculated as $∆V=∆\kappa v≅\pm 2\times 10^{-4}$ µm^3^. We report in Figure 7b the distribution of the calculated volumes for bMSC-4D and bMSC-10D depositions. In both cases, the error bar is well inside the data point’s dimension shown in the plot. Volume values are clearly very different in the two cases, being the average at 4 days of 0.09 µm^3^ and of 2.4 µm^3^ at 10 days. The volume corresponding to the deposition with a quasi HA spectrum in bMSC-4D is reported in blue. The average number of voxels in the z direction (depth) was used to estimate the average depositions thickness necessary to derive the concentration, as described in the next paragraph.

### 2.5. Ca Concentration in the Depositions

Concentration calculations were carried out following the work of Kahil and colleagues [11]. The integrated absorbance of the L_2_ peak obtained from the measured transmission across the Ca L_2_ edge is assumed to be proportional to the absorption cross-section, the thickness, and the number of Ca atoms in the selected depositions [25]. Measuring the average thickness for each deposition from the reconstructed volumes at 349.2 eV and using a calculated value for the cross-section, it is possible to obtain an estimation for the number of Ca atoms; i.e., the Ca content of the selected deposition. The procedure is reported in detail in the methods section. Concentration values for bMSC-4D and bMSC-10D distributes clearly in two distinct populations (Figure 7c). In particular, the Ca content in crystalline calcite is almost twice that of HA.

## 3. Discussion

Biomineralization is a ubiquitous and tightly regulated process by which living systems generate organized mineral crystals and in humans culminates with the formation of HA which is a naturally occurring mineral form of calcium apatite with the general formula (Ca10–*x*Y*x*)(PO4to)6(OH)_2_–*p*(CO_3_)*p*, where Y indicates the typical substituting metals (Zn, Mg, Sr) [26,27]. The process that explains the genesis and propagation in the extracellular matrix of the mineral remains largely unexplained and is highly controversial. In 2012, Boonrungsiman et al. [28] proposed a model by which amorphous calcium phosphate and ionic calcium stored in mitochondria are transported via vesicles to the extracellular matrix before converting to more crystalline apatite. Recently, Tang et al. [29] suggested that the process of biomineralization starts by carrying calcium and phosphorous clusters from the endoplasmic reticulum to mitochondria during bone formation. The authors found in the developing mouse parietal bone and dentine that the collagen mineralization gets started from the amorphous mineral phase evolving toward an ordered alignment of apatite. These two intriguing models on the early phase of the biomineralization have been formulated by the use of a murine model, while studies in human cell lines are still scarce. The present study is focused on the characterization of the intracellular mineral depositions in human bMSC induced to differentiate toward osteoblast after 4 and 10 days from the osteogenic induction. Our experiment, conducted in static 2D culture system, did not take into consideration the mechanical loading and niche extracellular matrix stiffness, crucial aspects for the osteblastic differentiation [30]. However, as shown in a previous work [30] using the same osteogenic differentiation protocol employed in the present study, we observed the overexpression of *RUNX2*, the master regulator of osteogenesis, and *SP7*, coding for Osterix, a key transcription factor required for osteoblast differentiation. CryoSXT at 520 eV was used to obtain the general ultrastructure description of cryopreserved bMSC, while cryoSXT and cryoXANES at the Ca L edges were combined to characterize their intracellular Ca-rich depositions, specifying their morphology, their distribution, and their Ca content chemical state. The method employed was already successfully applied in precedent studies on other cell types, with smaller dimensions and simpler internal structure [11,12,13]. Similar observations for the cells used in this study are more complicated because of their bigger transversal dimensions, bigger thickness and richness of organelles, vesicles, and dense structures in the cytoplasmic matrix (Figure 2). All the dense bodies localized in the intracellular matrix did not show any Ca content by spectroscopy analysis, and we did not reveal any membranes around the Ca-rich depositions.

Almost all the spectra extracted from the depositions after 4 days from the osteoblastic differentiation induction correspond to CaCO_3_, and they showed a ratio between the main L_3,2_ peaks (a_2_ and b_2_ in Figure 4b) and the corresponding pre-peaks (a_1_ and b_1_) compatible with the calcite measured reference sample. This indicates the same short-range order around the Ca atoms, which are very close to being octahedrally coordinated by oxygen atoms [22]. These results corroborate the hypothesis launched in a recent paper by Procopio et al. that the genesis of bone Ca depositions starts with a compound bound to carbonate [20]. Herein, besides confirming this hypothesis, we characterized the crystalline and molecular structure of intracellular depositions, finding that calcite is the precursor of HA. This result is somewhat surprising, as the crystal structure of the depositions forms very early in the biomineralization process (4 days from differentiation induction). We could speculate that in the initial phase of depositions (earlier than 4 days), the CaCO_3_ starts as an amorphous phase and then evolves to crystalline calcite. The striking finding is that we detected both crystalline compounds, calcite and hydroxyapatite—although the latter is rare—within the cell during the early phase of osteogenic differentiation (4 days), redefining most of the models of the biomineralization process published so far for this kind of cell [28,29]. Nevertheless, we cannot exclude that the biomineralization starts as ionic Ca within mitochondria in the earlier phases (before 4 days of differentiation).

In bMSC-10, all the Ca depositions analyzed showed similar absorption spectra that can be attributed to HA crystals suggesting that the process of biomineralization in differentiating bMSC starts within the cells with crystalline calcite (4 days) and culminates with the formation of hydroxyapatite crystal (10 days). This intriguing paradigm for the crystalline HA formation has been described in vitro by Marchegiani and colleagues [31]. Indeed, they formulated the hypothesis that a superficial dissolution of calcite promotes the overgrowth of hydroxyapatite. In this process, the biogenic calcite crystals have a role in recruiting calcium ions at the crystal surface. In addition, they observed that only the biogenic calcium carbonates were transformed to HA. This is because the biogenic calcium carbonates host an organic matrix that alters the crystalline structure of calcite. On the contrary, synthetic calcite crystals did not show the same behavior, because they are less soluble [32]. Moreover, the isomorphic substitution of magnesium to calcium ions also destabilizes the structure of calcite and enhances the solubility of biogenic crystals [33]. In a recent paper, we revealed in the early phase of biomineralization, different timing of bone elements accumulation, highlighting the key role of Zn. Indeed, we detected a first co-localization of Ca and Zn [20]. Zn and Mg are typical substituting metals of Ca in HA [26].

Since calcite acts as a source of calcium ions, they react with phosphate ion species, forming calcium phosphates, and the specific phase of calcium phosphate is controlled by the pH. This step, where the phosphate binds the Ca ions forming calcium–phosphates is unequivocally proven by the peculiar “hook” shape of the L_2_ pre-peak of the spectra shown in Figure 5c,d.

The depositions of HA measured in bMSC-10 showed a higher volume with respect to the deposition of calcite found in bMSC-4, suggesting a chemical and crystalline rearrangement where atoms of phosphorous and other substitutive elements (Mg and Zn) contribute to form hydroxyapatite minerals. It is worthy to note that the increase of volume in the depositions at 10 days after the osteoblastic induction corresponds to a decrease of Ca concentration (g/cm^3^) compared with depositions at 4 days. The decrease (about half) of Ca content in HA-rich depositions with respect to calcite-rich depositions could be ascribed to the different number of Ca atoms per unit cell volume of crystal lattice dimension of calcite with respect to HA being 4.1 × 10^−2^ and 1.9 × 10^−2^ (Ca/Ǻ^3^), respectively (http://ruby.colorado.edu/~smyth/min/apatite.html, accessed on 16 September 2020).

It must be underlined that calcium mineral nuclei are embedded in a dense pool rich in organic matrix. Indeed, as suggested by Addadi et al., the morphology and dimension of biogenic crystals may be modulated during crystal growth by controlled adsorption of ionic, molecular, or macromolecular additives, which are often eventually occluded inside the growing crystal [1]. The occlusion of these additives increases the volume of the composite depositions, which are sometimes referred to as a biogenic mineral or biomineral, diluting the calcium content.

## 4. Materials and Methods

### 4.1. Isolation and Culture of Human Mesenchymal Stem Cells

Human mesenchymal stem cells were isolated from the bone marrow of healthy male volunteers (Prof. Berti, Policlinico, Milan). The informed consent from all the subjects has been obtained in compliance with the Helsinki declaration according to institutional guidelines and regulations of the Ethical Committee of “IRCCS Policlinico” Milano. Unexpected or unusually high safety hazards have been encountered. bMSCs have been tested for purity by flow cytometry and cultured in Dulbecco’s modified Eagle’s medium with 1000 mg/L glucose, 10% fetal bovine serum (FBS), and 2 mM l-glutamine (culture medium) at 37 °C. When confluent, the cells were detached by treatment with trypsin–EDTA 1× (Sigma–Aldrich), characterized, subcultured, and used at passage numbers 3–5 [34].

### 4.2. Osteogenic Differentiation of bMSCs

To induce osteogenic differentiation, bMSCs were seeded in 96-well plates. Once the cells were confluent, an osteogenic cocktail was added, containing 2 × 10^–8^ M 1α,25-dihydroxyvitamin D3, 10 mM β-glycerolphosphate, and 0.05 mM ascorbic acid (Sigma–Aldrich) [35]. To investigate calcium deposition in differentiated bMSCs, the cells were rinsed with PBS, fixed (70% ethanol, 1 h), and stained for 10 min with 2% Alizarin Red S (pH 4.2, Sigma–Aldrich) [35]. The experiment was repeated three times in triplicate. Alizarin Red S staining was released from the cell matrix by incubation in 10% cetylpyridinium chloride (Sigma–Aldrich) in 10 mM sodium phosphate (pH 7.0), for 15 min, and the absorbance was measured at 562 nm.

### 4.3. Sample Preparation for Synchrotron-Based Techniques

BMSCs were seeded onto gold quantifoil R 2/2 holey carbon-film microscopy grids. The cells were plated at a concentration of 1 × 10^4^ cell/cm^2^ on the grids previously sterilized. After 4 and 10 days from the osteogenic induction, the attachment and spreading of the cells was carefully verified using visible light microscopy. Cell culture medium was removed, and the membranes were briefly washed in 100 mM freshly prepared ammonium acetate solution two times to remove salts and trace metals from the medium. The cells were frozen–hydrated by a rapid plunge freezing in a liquid ethane pool cooled with liquid nitrogen with a Leica EM GP. Excess water was carefully removed before plunge freezing via 2 s blotting to avoid the formation of absorbing vitreous ice upon the samples. Under cryogenic conditions, the frozen specimens were transferred into the Mistral transmission X-ray microscope [36,37]. Calcite references samples were prepared by finely crushing calcite powders (Sigma–Aldrich) in a mortar, and the obtained dust was laid down on a Quantifoil Au TEM grid. The same procedure has been used for the preparation of HA reference samples (provided Bio Eco Active S.R.L, Bologna, Italy).

### 4.4. Cryo-Soft-X-ray Tomography

CryoSXT images were recorded at the MISTRAL beamline of the ALBA Synchrotron, where photons extracted from a bending magnet source are focalized onto the sample by a capillary condenser placed in front of the monochromator exit slit. After the sample, a Fresnel zone plate with an outermost zone width of 40 nm acts as an objective lens of the microscope, generating a magnified image on a direct illumination CCD detector located a few meters from the sample [36,37]. CryoSXT was carried out at 349.2 eV and 520 eV to optimize the contrast between the calcium- and carbon-rich objects and the surrounding water-rich cytoplasmic solution without staining, sectioning, or using enhancing agents. Cryogenic conditions were maintained during all the experiments. For each cell, a tilt series was acquired using an angular step of 1° on a 110° angular range. The effective pixel size in the images was 13 nm at 349.2 eV and 17.3 nm (to image the highest possible field of view, although at a resolution cost) at 520 eV. No radiation damage was detected at the achievable spatial resolution. Each transmission projection image of the tilt series was normalized using flat-field images of 1 s acquisition time. The tilt series were manually aligned using eTomo in the IMOD tomography software suite [38]. Au fiducials of 150 nm from BBI solution were used for projection alignment purposes. The transmission tilt series were finally reconstructed with TOMO3D [39], using the SIRT iterative-algorithm with 30 iterations and then segmented by Amira (Thermo Fisher Scientific, Waltham, MA, USA).

### 4.5. Spectromicroscopy

CryoXANES or spectromicroscopy was also performed at the MISTRAL beamline of the ALBA synchrotron. Scanning the X-ray energy through the Ca L edge, cryoXANES can be used to determine the Ca bulk chemical state with spatial resolution of few tens of nanometers, providing pixel-by-pixel absorption spectra over a 13 μm by 13 μm field of view [12]. Then, 2D Ca L edge XANES images were collected on representative areas (12 s exposure time) using an effective pixel size of 13 nm and with a variable energy step (0.5 eV of pre-edge and post-edge, 0.1 elsewhere). Monchromator slits gaps and constant focus constant values were chosen in order to optimize the compromise between energy resolution and contrast in the images. The necessary total acquisition time was about 1.5 h per energy scan, including the flat field acquisition at each energy step.

The transmitted intensity at each energy value is normalized to unity dividing by the corresponding flat field image, i.e., the incident intensity on the sample, taking into account the value of the electron current in the storage ring. Then, all the transmission images are aligned with respect to the first image, applying the x–y shifts, which maximize the cross-correlation between the same selected ROI in the two images. These shifts are calculated using the Python library of cv2 “Open Computer Vision” (the used function is “cv2.matchTemplate()”). Usually, this operation reduces the effective dimension of the field of view by 10 to 15%. Finally, the absorbance for each pixel can be calculated from the measured transmission as:(1)$$A=\mu t=-ln\left(\frac{I}{I_{0}}\right)$$
and can be related as a function of the energy to the linear absorption coefficient using the Beer–Lambert law:(2)$$T\left(x,y\right)=\frac{I\left(x,y\right)}{I_{0}\left(x,y\right)}=e^{\left(-\int \mu \left(x,y,E0\right)dt\right)}=e^{\left(-\int {µ}_{m}\left(x,y,E0\right)\rho dt_{m}\right)}$$
where *t* is the thickness, *μ_l_* is the linear absorption coefficient of the material, and *μ_m_* is the mass absorption coefficient.

In each intensity measurements also an unknown background is acquired, which is few percent of the incident intensity. As a consequence, the spectral shape may appear distorted with essentially the high-intensity peaks appearing flattened [40]. Indeed, even if they span a reasonable range of values, the absolute values for the concentration reported could be systematically underestimated for this reason. Nevertheless, because of the relatively small thickness of the Ca depositions, peaks ratio in the same deposition shows good linearity and measured spectra don’t appear distorted respect with the measured references.

Spectra were extracted only from pixels that satisfy the following signal to noise criteria:(3)$$\Delta \mu_{l}t=\left[\mu_{l}\left(349.2eV\right)-\mu_{l}\left(342eV\right)\right]t>2N$$
with *µ_l_* as the linear absorption coefficient, *t* as the thickness, and N as the noise defined as the absorbance standard deviation in the pre-edge energy region (341–343.5 eV).

The pixel selection operation was performed using a homemade function created in Matlab.

Peak fitting was performed to compare the measured spectra and in particular to estimate the height and the area of L_2_ and L_3_. To eliminate the contribution of lower energy absorption, the absorbance values in the pre-edge energy region were linearly fitted using the Matlab function “polyfit”, and the corresponding best fitting straight line was subtracted on the full energy range. Then, in order to eliminate the edge step arising from atomic transitions to the continuous, a double arctan function was also subtracted, following the procedure described in [41]. In this case, the Matlab function “CreateFit” was used. The used double arctan function was:(4)$$\frac{h_{1}}{\pi }\left\{atan\left[\frac{\pi }{w_{1}}\left(E-E_{1}\right)\right]+\frac{\pi }{2}\right\}+\frac{h_{2}}{\pi }\left\{atan\left[\frac{\pi }{w_{2}}\left(E-E_{2}\right)\right]+\frac{\pi }{2}\right\}+C$$
with *w*_1_ = *w*_2_ = 0.2 eV, *E*_1_ = L2peakmax, and *E*_2_ = L2peakmax + 3 eV. The other parameters (*h*_1_, *h*_2_, and *C*) were fitted with the data points in the pre-edge and in the post-edge (355.8–360 eV) energy regions.

Afterwards, all the peaks present in each spectrum were fitted by Gaussians (the chosen number of Gaussians corresponds to the number of peaks), creating a fitting model in Larch software [42]. An example of fitting results for a calcite spectrum is reported in Figure 8.

### 4.6. Ca Concentration Estimation

Concentration calculations were carried out assuming linearity between absorbance, *A*, and Ca concentration using:(5)$$A=\mu_{l}\cdot t=n\cdot \sigma \cdot t\Rightarrow n=\frac{A}{\sigma \cdot t}$$
where *A* is derived integrating the intensity over the L_2_ peak by the software Larch, *μ_l_* is the linear absorption coefficient, and ***t*** is the thickness. The linear absorption coefficient was written using the absorption cross-section *σ*, which is theoretically estimated to be *σ* = 0.0625 Å^2^ per atom [11], and the number of atoms of Ca atoms per unit of volume n, which is the concentration value we estimated in g/cm^3^ in Figure 7c using the Ca atomic weight (40.078 u). The volume of each deposition was calculated by the use of the reconstructed volume from cryoSXT at the Ca L_3_ edge. In particular, each deposition has been segmented fixing a threshold using Fiji software [43]. From the resulting three-dimensional bitmap, we calculated the deposition total thickness and the corresponding error, considering an isotropic pixel of 13 × 13 × 13 nm^3^. The thickness error result was small enough to be neglected; then, the final uncertainty on the Ca concentration ∆[Ca] was calculated as follows:(6)$$\Delta \left[Ca\right]=\frac{\Delta A}{A}\times \left[Ca\right]$$

### 4.7. Transmission Electron Microscopy (TEM)

For ultrastructure analysis, cell pellets were fixed overnight in 2.5% glutaraldehyde in 0.1 M sodium cacodylate buffer (pH 7.4). Samples were post-fixed with 1% osmium tetroxide for 90 min, dehydrated, and embedded in Epon-Araldite resin. Ultrathin sections, obtained with a Leica Supernova ultramicrotome (Reichert Ultracut E and UC7; Leica Microsystems, Wetzlar, Germany), were stained with lead citrate and observed with a Zeiss EM10 electron microscope (Carl Zeiss, Oberkochen, Germany).

## 5. Conclusions

The main conclusion of the study is that in the process of biomineralization, a precursor of mature hydroxyapatite is calcite: a carbonate compound of calcium. This finding is consistent with the hypothesis launched in a previous study showing that the genesis of bone Ca depositions starts with compound not bound to phosphate [20]. We may speculate that in the initial phase of depositions (earlier than 4 days), the CaCO_3_ starts as amorphous phase and then evolves to crystalline calcite. Moreover, we detected both crystalline compounds, calcite and hydroxyapatite, during the early phase of osteogenic differentiation (4 days), although a lesser amount of hydroxyapatite is present at this stage. We propose in Figure 9 model of early phases of biomineralization combining our present and previous results with the results and hypothesis taken from other studies. Other actors are in the play include magnesium, zinc, and intracellular vesicles, and although there are several hypotheses on their roles, the moment of their entry on the scene remains elusive.

## Figures and Tables

**Figure 1 ijms-22-04939-f001:**
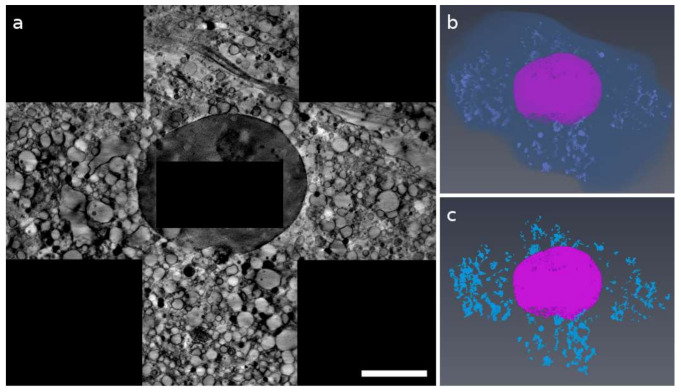
(**a**) Central slice of a bMSC-4D cell volume collected at 520 eV and obtained by stitching together four tomographic reconstructions. Scale bar is 6.9 μm. (**b**,**c**) Corresponding color-coded 3D rendering (with the nucleus in pink and the dense objects inside the cell in light blue). These objects are different in shape and dimensions and were automatically selected using a threshold. In (**b**), the cellular membrane is also showed.

**Figure 2 ijms-22-04939-f002:**
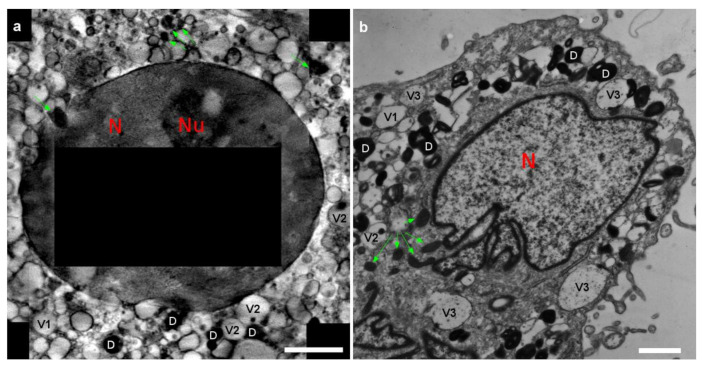
(**a**) bMSC-4D detail from the cryoSXT reconstruction slice reported in Figure 1a with letters and arrows indicating different objects inside the cell. Nucleus (N), mitochondria (green arrows), lipid droplets, vesicles (V1 and V2), and dense objects (D). Scale bar is 3.4 μm. (**b**) bMSC-4D TEM image. Despite the different sample preparation (cryo vs. dried and stained) and the different thicknesses tackled with each technique, we can still recognize the same objects indicated in (**a**) (same labeling was used for V1 and V2, V3 which represent vesicles containing smaller dense objects). It is possible to notice that the contrast in 2a is quantitative (related to the elements contained in each tomogram voxel), while the TEM image is not as the contrast is produced by the staining. In addition, the global shapes are better preserved in the cryo preparation (e.g., nucleus). Scale bar is 1 μm.

**Figure 3 ijms-22-04939-f003:**
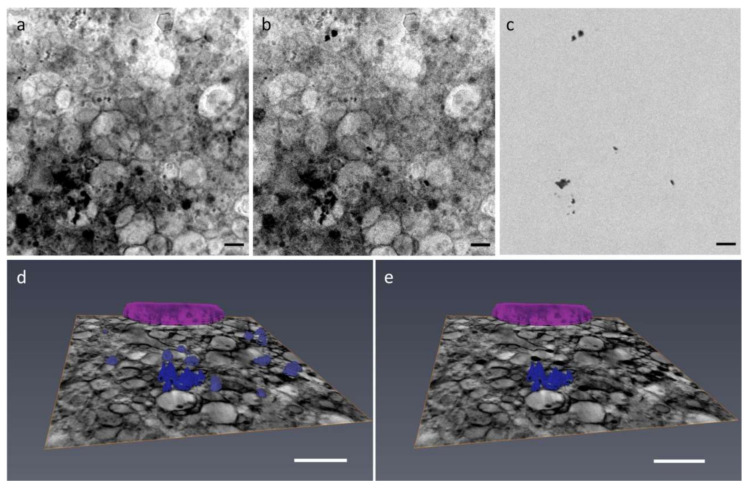
(**a**,**b**) Average of absorbance projections of a cytoplasm region in the cell showed in Figure 1 (≈10 μm × 10 μm) recorded at the pre Ca-edge energy region (≈342 eV) and at the Ca L_3_ peak maxima (≈349 eV), where the contrast between Ca and the other elements is maximized. (**c**) Image difference between (**b**) and (**a**). Since the absorbance of all the other elements is almost the same in (**a**,**b**), their difference (**c**) reveals the Ca 2D distribution in the field of view. Scale bar is 1.3 μm. (**d**) Color-coded 3D rendering of the nucleus and dense structures of another bMSC-4D. Dense structures voxels were selected automatically using a threshold. After spectral analysis, only some of these dense structures show Ca content. These Ca-rich voxels are reported in (**e**). Scale bar is 2 μm.

**Figure 4 ijms-22-04939-f004:**
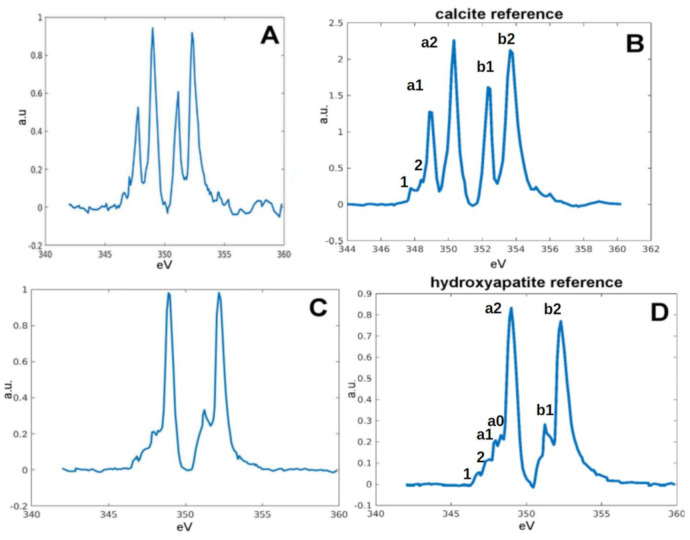
(**a**) Calcite spectra detected in bMSC-4D. (**b**) Characteristic spectrum of calcite measured in a reference standard. (**c**) HA spectra detected in bMSC-10D. (**d**) Characteristic spectrum of HA measured in a reference standard. In (**b**,**d**), the main spectral features are indicated by letters and numbers.

**Figure 5 ijms-22-04939-f005:**
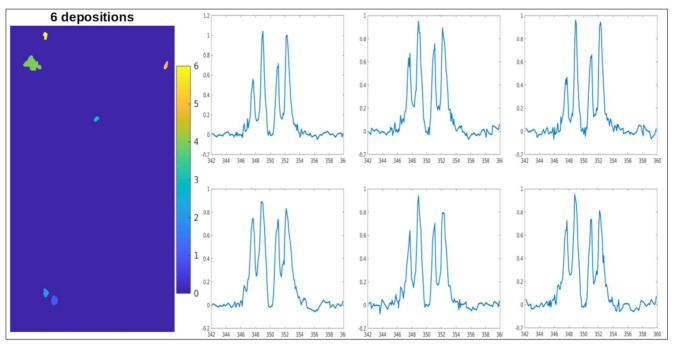
(**a**) Binary map with the Ca depositions of the field of view in Figure 3a–c from bMSC-4D. Colored pixels are the ones that satisfy the S/N criterion defined in the text. Six depositions were big enough to allow spectroscopic analysis, and the respective spectra are reported from panel (**b**–**g**) with the number of pixels of the corresponding deposition. The showed spectra have peaks structure and relative peaks intensities similar to the calcite reference standard (Figure 4b).

**Figure 6 ijms-22-04939-f006:**
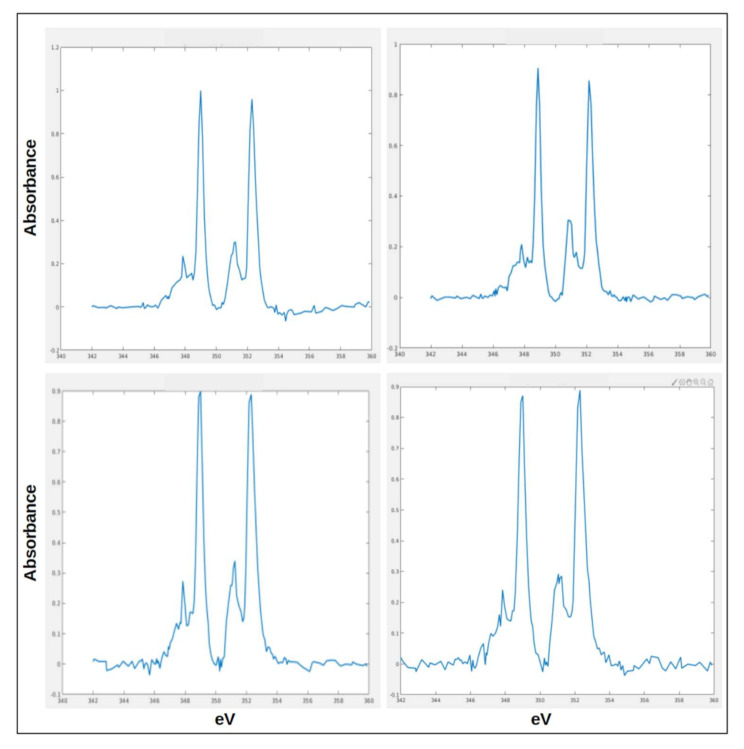
Spectra from depositions in bMSC-4D that show features similar to HA reference spectrum. These four spectra are the only spectra found in bMSC-4D which cannot be attributed to calcite. Despite the low signal-to-noise ratio, they show indeed features similar to the ones observed in HA spectra from bMSC-10D.

**Figure 7 ijms-22-04939-f007:**
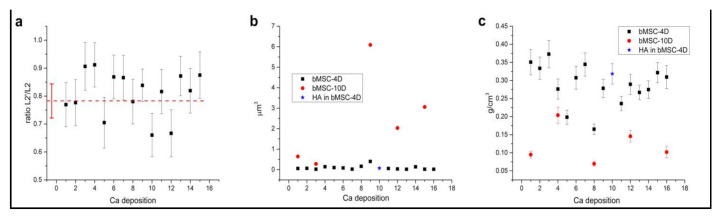
(**a**) Ratio values of the pre-peak L_2′_ to the L_2_ peak (L2′/L2) calculated for Ca carbonates-rich depositions in bMSC-4D. These values are compared with the one obtained from pure biological calcite reference spectrum (dashed line in red). All the measured ratios are compatible within the error with the one estimated from the reference sample, strongly supporting the hypothesis of presence in the observed depositions of crystalline calcite crystals. (**b**) Ca depositions volume values in bMSC obtained using the reconstructed volumes from cryoSXT at the Ca L_3_ edge. (**c**) Ca content in g/cm^3^ calculated for the depositions characterized with cryoSXT at Ca L edge.

**Figure 8 ijms-22-04939-f008:**
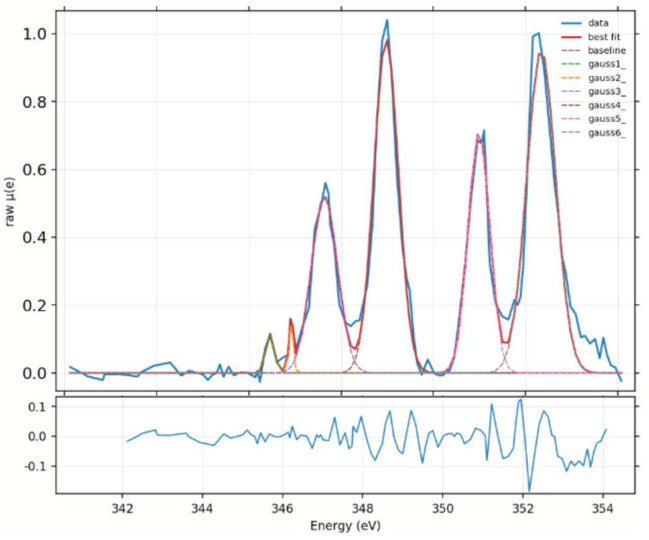
Gaussian peaks fit shown for a typical measured calcite spectrum. The number of fitted peaks is six and they have similar FWHM. The fit result was used to calculate the data reported in Figure 7a,c as explained in the corresponding methods sections. The residual, calculated simply as the difference between the fit and the experimental data, is also reported in the inset below the fit.

**Figure 9 ijms-22-04939-f009:**
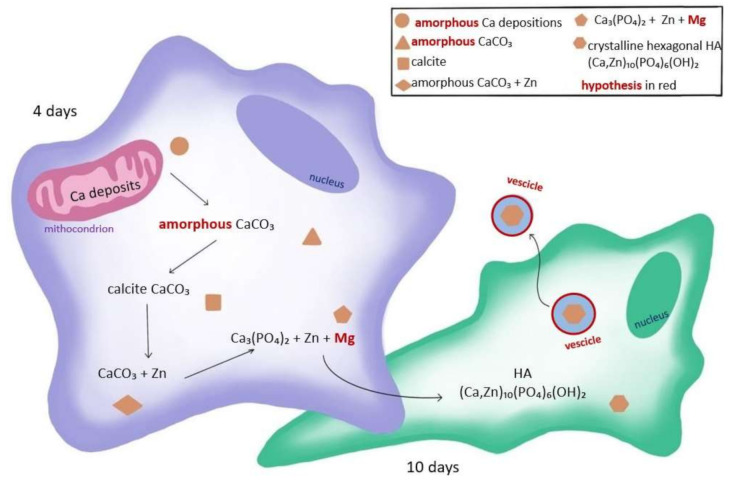
Model of early phases of biomineralization based on the result of this study and results and hypotheses taken from other studies [20,28,29] showing the localization and composition evolution of Ca compounds during the early phases of osteogenic differentiation. For the sake of simplicity, we did not report the transportation of Ca deposition from endoplasmic reticulum to mitochondria, as proposed by Tang et al. [29].

## Data Availability

The data presented in this study are available in article.

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
