# Peer review of "Calcite as a Precursor of Hydroxyapatite in the Early Biomineralization of Differentiating Human Bone-Marrow Mesenchymal Stem Cells"

_ijms, 2021, doi:10.3390/ijms22094939_

Round 1
Reviewer 1 Report
Dear Authors:
It was a pleasure to read your manuscript. I have some minor comments.
1) The biomineralization depends on stresses/strain (mechanical loading) a lot. Discuss it.
2) Amorphous Ca-P-O-H phases and concentration gradients may play an essential role. Include some comments.
3) The CaCO3 just may be a by-product of an amorphous phase which is the real nucleus before crystalline islands of HA. Discuss it bit more.
4) pH-Value changes are a way in biomineralization for regulating the growth of boen. Include some comments.
5) Natural bone itself is a mixture of amorphous and defective crystalline HA. Include comments in the paper.
It was nice to learn about the remarkable versatility of synchrotron- 65
based cryo-soft-X-rays transmission microscopy techniques.
Sincerely,
Yours reviewer
Reviewer 2 Report
Review:
The authors have provided a manuscript on the presence of Calcium Carbonate (Calcite) as a possible precursor to Hydroxyapatite depositions and biomineralization found in the ECM of bone cells (bMSC). I commend the authors on a good manuscript and clear presentation of data that was easy to read and follow. I have a few comments and suggestions that I believe will enhance the appreciation of the readership of this paper.
- In the introduction, you mention using CryoSXT and CryoXANES microscopy (pg 1, lines 69-72). It may help to add a few sentences about how the technique works since your manuscript will have appeal to a group with limited knowledge on them.
- You mention that the resolution of these techniques is “few tens of nanometers” (pg 1, line 72). How sure are you that you are able to pick up all the precursor Ca sites?
- You have mentioned that past analyses has focused on more advanced stages of bone mineralization and your investigation on day 4 post induction is novel. Can you talk more about this pick? Why did you pick day 4 instead of a range of days up to 10?
- Your key finding of the presence of Calcite is interesting. However, you also find HA. Do you think going for an earlier day post-induction would have resulted in only Calcite findings? Asked a different way, how sure are you that the Calcite you find is the precursor to HA? You also refer to the data in Figure 6 showing HA spectra as peculiar in the figure legend. May I know why you consider this peculiar?
- Your images and analyses are impressive. How representative are they? Would you mind providing information on how many samples were analyzed in each condition? I ask this because even slight changes in induction conditions result in large consequences downstream.
- I’m not an expert on this technique and hence I have a question of what happens between figures 3d and 3e. If I understood correctly, Calcium rich clusters were identified using Calcium peaks (349eV-342eV). However, spectral analysis resulted in a bunch of the clusters being removed from analysis since they did not contain Ca. How is this possible? Does this suggest that Calcium peaks (349eV-342eV) being used are not a reliable measure of Ca?
Smaller concerns:
- Figure 2: A has V1 and V2 but B has V1 and V3. V3 is not talked about in the legend.
- Page 4 line 126 Add (micro)
- Figure 4 legend: line 187 b -> c
- Figure 6 y-axis: change assorbance -> absorbance
Round 2
Reviewer 1 Report
Dear Authors:
The paper has a high quality in writing style and descibe the research methods.
However, one needs to consider:
For biominerlization the mechanical loading has an important influence. This point is not considered in the paper appropriate. Therefore, the research design could be improved. One can not conclude straigth forward from the in-vitro experiements to the real conditions in streamed through and mechanical loaded in-vitro situation. Mayby, stem cells do behave very different in this hostile and stressed situation. They need more time to aclimate.
Best regards
Reviewer
Author Response
We thank the reviewer for the suggestion and we add a comment on the mechanical load in the discussion.
Reviewer 2 Report
I thank the authors for addressing all my comments. As a minor point, I suggest the authors include a reference for Pt. 3 to strengthen their case.
Author Response
We investigated early phase of biominarilaziotion and since in a previous study (ACS) at four days we found that the mineral depositions did not contained P, we speculated that they could have been CaCO3. This is why we focussed on 4 days finding that our speculation was correct and surprisingly that the CaCO3 was crystalline (calcite). In fact, in the added figure we update the prevous model reported in ACS to include the present findings.